# Burden of Mental and Behavioral Disorders in Colombia, 2022: A Subnational Analysis Based on Disability-Adjusted Life Years

**DOI:** 10.3390/ijerph22121854

**Published:** 2025-12-12

**Authors:** Karen Julieth Quintero Díaz, Oscar Alexander Gutierrez Lesmes, Emilce Salamanca Ramos

**Affiliations:** 1School of Public Health, Universidad de los Llanos, Villavicencio 500001, Meta, Colombia; oagutierrez@unillanos.edu.co; 2Gesi Research Group, School of Public Health, Universidad de los Llanos, Villavicencio 500001, Meta, Colombia; esalamanca@unillanos.edu.co

**Keywords:** global burden of disease, mental disorders, disability-adjusted life years, premature mortality, public health

## Abstract

Mental disorders encompass conditions that affect cognition, emotions, and behavior, representing a major public health challenge. In Colombia, there are no studies that estimate the burden of disease caused by mental and behavioral disorders. This study aimed to determine the burden of disease attributable to these conditions in the departments of Colombia in 2022. A burden of disease analysis was conducted using official national data sources, including the Individual Health Service Delivery Records and death certificates from the Vital Statistics System, consolidated by the Ministry of Health and Social Protection within the Integrated Social Protection Information System. Estimation methods followed the World Health Organization’s Global Health Estimates framework. Disability-Adjusted Life Years were used as the summary measure, integrating mortality and non-fatal outcomes to quantify the overall population impact. A total of 296,010.6 Disability-Adjusted Life Years were estimated (95% UI: 279,343.2–312,678), representing a rate of 572.7 (95% UI: 540.5–605) per 100,000 population. Anxiety accounted for 47.26%. Women represented 60.86% of the total burden, with 180,157.6 (95% UI: 165,046.3–195,268.9). Overall, 99.27% of the burden from mental and behavioral disorders was due to Years Lived with Disability, underscoring the substantial impact on quality of life, particularly among women.

## 1. Introduction

Mental disorders comprise a set of clinically identifiable symptoms and behaviors, cognition, emotional regulation, and behavioral functioning leading to significant limitations in daily activities [1,2,3]. In recent decades, mental and behavioral disorders (MBDs) have become an increasing public health priority due to their high prevalence and substantial impact on quality of life, healthcare systems, and national economies [4,5].

Accurately assessing their population impact requires metrics that integrate both mortality and disability. Traditional indicators based solely on mortality or prevalence fail to capture the duration, severity, and functional consequences of these conditions. Disability-Adjusted Life Years (DALYs), used in the Global Burden of Disease (GBD) framework and adopted by the WHO Global Health Estimates (GHE), address these limitations by combining Years of Life Lost (YLL) and Years Lived with Disability (YLD) into a single measure [6,7,8,9,10].

In Colombia, population-based assessments of mental and behavioral disorders have relied predominantly on mortality and prevalence statistics, limiting the understanding of their functional impact. Although DALYs have become essential worldwide for guiding health policies and prioritizing interventions, their use in Colombia remains limited, and no recent subnational studies have estimated the burden of disease associated with MBDs using GBD methodology [6,11,12,13].

This gap restricts the capacity of territorial health authorities to plan and allocate resources based on comparable and standardized burden estimates. Therefore, the present study aimed to quantify the burden of disease attributable to mental and behavioral disorders across Colombian departments in 2022, providing updated evidence to inform national and subnational decision-making.

## 2. Materials and Methods

A descriptive study was conducted using secondary data sources. Estimation procedures were based on the creation of synthetic indicators, such as YLL, YLD, and DALYs. The study population consisted of Colombians in 2022, according to records from the National Administrative Department of Statistics (DANE), totaling 51,874,024 people (distributed across the country’s 32 departments). From this population, individuals who became ill or died from mental or behavioral disorders during that period and were properly registered in official databases were selected. Mortality data from Colombia’s Vital Statistics System were obtained through the Single Affiliation Registry (RUAF) database, and the Individual Health Service Provision Records (RIPS), both of which are available in the respective databases of the Integrated Social Protection Information System (SISPRO). They have restricted access that requires a username and password assigned by the Ministry of Health and Social Protection (MSPS), in accordance with Law 1581 of 2012 [14].

Disorders were classified according to their frequency of occurrence. Morbidity causes were identified using the primary diagnosis, and mortality causes through the underlying cause of death. Records lacking primary diagnoses or underlying causes, as well as duplicate entries, were excluded. Organic mental disorders, including symptomatic disorders, mental and behavioral disorders caused by tobacco use, mental and behavioral disorders derived from the use of other stimulants including caffeine and unspecified mental disorders, were also omitted. Likewise, the study did not incorporate diseases with no assigned weights in the methodology, as well as the group labeled as “others” or “unidentified.” These diseases did not allow for clear classification and weight assignment within each group.

A total of 112 ICD-10 codes were selected, each associated with an estimated disability weight drawn from the WHO (Appendix A). Table of sequelae and health states. The WHO Global Health Estimates framework was applied, disaggregated by sex and age group, and disability weights for mental and behavioral disorders were assigned following this methodology [15]. YLL and YLD were calculated using the WHO’s abridged formula, incorporating life expectancy, number of cases, and estimated duration of disability. DALYs were obtained by summing YLL and YLD by age, sex, and department (Appendix A).

Although administrative health databases provide wide population coverage, they are subject to potential underreporting, miscoding, and variability in diagnostic practices, which were considered when interpreting the results.

Equation (1): Basic formula for YLLs.(1)$${YLL}_{c,a,s,t}=D_{c,a,s,t}e_{x}^{*}$$

*D* is the number of deaths due to the cause (*_c_*) in the age group (*_a_*), in sex (*_s_*), and year (*_t_*) $e_{x}^{*}$ is the life expectancy at each age (the weighting factor is derived from the standard life expectancy (SLE) recommended by WHO, based on a 92-year-old SLE).

Equation (2): Basic formula for YLDs.(2)$${YLD}_{c,a,s,t}=W_{c}*\;P_{c,a,s,t}$$

*W* is the disability weight, *P* is the prevalence of the disease or injury (*_c_*), in the age group (*_a_*), according to sex (*_s_*), and year (*_t_*), according to the Global Burden of Disease Study [15] disability weights for each health state.

Equation (3): Basic formula for DALYs.(3)$${DALY}_{c,s,a,t}={YLD}_{c,a,s,t}+{YLL}_{c,a,s,t}$$

(*_c_*) is disease or injury, (*_a_*) the age group, (*_s_*) the sex, and (*_t_*) is the year.

* The study presents crude rates that were not age-standardized

The indicators were calculated using IBM SPSS Statistics for Windows, Version 23.0 (IBM Corp., Armonk, NY, USA).

Ethical considerations: The study was classified as minimal risk and conducted in accordance with the ethical principles established by national and international regulations. It followed the GATHER statement (Appendix A), which aims to enhance transparency in the reporting of health estimates based on multiple data sources [16]. Additionally, this study complied with the ethical guidelines for biomedical research involving humans established by the Council for International Organizations of Medical Sciences (CIOMS), specifically guidelines 1, 9, and 12 [17].

## 3. Results

### 3.1. Years of Life Lost (YLLs)

During 2022, a total of 59 deaths attributed to mental and behavioral disorders were reported in the country. Of these, 89.8% (53 cases) were related to disorders due to psychoactive substance use. A total of 2132.7 (95% UI: 1504.3–2761.1) YLL were calculated and attributed to mental and behavioral disorders, with a rate of 4.1 (95% UI: 2.9–5.3) YLL per 100,000 inhabitants. Regarding sex comparison, the premature death rate in men is 8.33 times higher than in women. Concerning age distribution, a higher number of YLL were observed in men over 60 years old, especially in the 60 to 64-year age group.

When classifying departments with the highest YLL rates, the department of Amazonas showed a rate of 42.5 (95% UI: 0–28.8) premature deaths, followed by Chocó with a rate of 12.2 (95% UI: 0–130.1) (see Figure 1) (Appendix A).

### 3.2. Years Lived with Disability (YLDs)

A total of 38,052 cases of mental and behavioral disorders (classified under 112 ICD-10 codes) were recorded. Of these, 19,453 occurred in men (51%). Regarding health loss, an estimated 293,877.8 (95% UI: 277,218.5–310,537.2) YLDs were calculated, with a rate of 568.6 (95% UI: 536.4–600.9) YLDs per 100,000 inhabitants (see Figure 2). As for sex, men had a 0.34 times lower rate of suboptimal health compared to women.

In the distribution of YLD rates by department, Quindío showed the highest YLD rate of 1329.3 (95% UI: 946.6–1711.9), attributable to mental and behavioral disorders. Meanwhile, Amazon and Orinoquía regions exhibited the lowest health loss values. The departments of Guainía 121.2 (95% UI: 73.2–168.3), Vaupés 101.4 (95% UI: 61.5–137.6), and Vichada 75.1 (95% UI: 51.1–98.8) stood out among them (Appendix A).

### 3.3. Disability-Adjusted Life Years (DALYs)

A total of 296,010.6 Disability-Adjusted Life Years were estimated (95% UI: 279,343.2–312,678), with a rate of 572.7 (95% UI: 540.5–605) DALYs per 100,000 inhabitants. This is the result of adding YLL and DALYs caused by mental and behavioral disorders. It represents the total DALYs produced by these disorders in Colombia for 2022.

At the departmental level, the highest burden was observed in neurotic disorders, stress-related disorders, and somatoform disorders, which accounted for 156,099.59 (95% UI: 140,954.4–171,244.8) DALYs. Schizophrenia, schizotypal disorders, and delusional disorders followed with 53348.6 (95% UI: 48,804.2–57,893.0). Finally, mood disorders accounted for 53,307.2 (95% UI: 51,182.7–55,431.5), (see Figure 3).

Quindío had the highest rate, with 1329.3 (95% UI: 946.6–1711.9). Risaralda followed with 1047.6 (95% UI: 715.8–1359.5), and Caldas, with 1006.6 (95% UI: 714.7–1297.9) (Appendix A). In age distribution, the burden increased with age, with patterns varying by specific disorder (Figure 4).

## 4. Discussion

In Colombia, in the year 2022, mental and behavioral disorders accounted for 296,010.6 DALYs (95% UI: 279,343.2–312,678), representing a rate of 572.7 per 100,000 population (95% UI: 540.5–605), of which 99.27% was attributable to YLDs. The marked predominance of YLD over YLL is consistent with international reports, where the burden of mental disorders arises primarily from prolonged disability rather than direct mortality [18,19,20]. This pattern is explained by the fact that most individuals with these disorders do not die from the primary condition but from other medical and behavioral comorbidities [7,21,22].

The study identified sex differences, with women contributing 180,157.6 DALYs (95% UI: 165,046.3–195,268.9), at rates of 681.1 (95% UI: 623.9–738.2), equivalent to 60.86% of the national burden from these disorders. This demonstrates greater vulnerability in this population group, consistent with recent international evidence showing a higher burden and prevalence of mental disorders among women [23].

In the analysis by specific cause, anxiety disorders accounted for 52.7% of total DALYs, followed by schizophrenia (18%) and bipolar disorder (8.9%). This order is partially consistent with global studies such as the GBD 2019 publication, where depressive disorders represented the largest proportion (37.3%), followed by anxiety (22.9%) and schizophrenia (12.2%) [18,24]. Differences may be justified by subnational variations in prevalence, diagnostic patterns, data availability, and territorial inequalities [25].

Regarding anxiety, this study estimated 156,099.6 DALYs (95% UI: 140,954.4–171,244.8), with a rate of 302.0 (95% UI: 272.7–331.3), which is consistent with global data. Studies report that anxiety affected nearly 46 million people worldwide in 2017 [26], and in 2019, the WHO reported approximately 301 million individuals living with an anxiety disorder [3], representing a large share of the global burden [27,28,29] Sex differences were evident, with women accounting for 62% of the burden, aligning with several studies [29,30]. With respect to age, a marked increase was observed among women older than 50 years, consistent with evidence linking biological aging and psychosocial changes to greater susceptibility to anxiety [31,32].

For schizophrenia, 53,348.6 DALYs (95% UI: 48,804.2–57,893.0) were recorded, with a rate of 103.2 (95% UI: 94.4–112.0). Males accounted for 59.1% of this burden, consistent with findings documenting a higher impact in this group [33,34]. The highest burden was observed among individuals aged 15–19 years, consistent with the typical age of onset described for this disorder [35,36,37].

Bipolar disorders caused 26,454.8 DALYs (95% UI: 25,1.00–27,852.6), with rates of 51.2 (95% UI: 48.5–53.9). Several studies report similar results [38,39], with more than 40 million people affected worldwide [3]. The burden among women reached 62.3%, although evidence is variable regarding which sex is most affected; some reports indicate that women have higher rates of bipolar disorders [39,40]. The 2019 GBD Mental Health study identified 39.5 million cases, of which approximately 52.4% were women [18].

Mental and behavioral disorders due to psychoactive substance use accounted for 13,046.74 DALYs (95% UI: 11,993.8–14,099.6), with a rate of 25.24 (95% UI: 23.2–27.3). A pronounced sex difference was identified, with males representing 79.1% of this burden, and a higher impact was observed among adolescents and young adults. These findings are consistent with studies identifying this group as the most vulnerable [41,42,43,44].

When disaggregated by substance type, alcohol accounted for 40.9% of DALYs attributable to psychoactive substances and 1.8% of total DALYs from mental disorders, while opioids represented 0.9%. These results reflect a greater burden from alcohol consumption in Colombia and align with GBD 2016 estimates, which reported approximately 4.2% of global DALYs attributable to alcohol and about 1.3% associated with psychoactive substances, including opioids [45]. Among the burden attributable to alcohol, 84% corresponded to males, a trend observed in Latin American studies and in GBD 2016 and 2021 estimates, where men are the major contributors to alcohol-attributable DALYs [40,45,46,47]. Additionally, in Colombia, the 2013 National Consumption Survey reported that 73% of alcohol users were men [48].

Territorially, three departments showed the highest rates: Quindío with 1329.3 (95% UI: 946.6–1711.9), Risaralda with 1047.6 (95% UI: 715.8–1359.5), and Caldas with 1006.6 (95% UI: 714.7–1297.9). In all departments, anxiety was the main contributor to DALYs, except in Magdalena, where schizophrenia reached a rate of 224.2 (95% UI: 157.3–291.1), surpassing other disorders in that territory. These differences may be explained by a combination of structural factors such as greater urbanization, population concentration, and better access to mental health services, which facilitate detection. In territories with greater limitations, such as Magdalena, severe disorders may represent a larger burden due to limited availability of specialized services, diagnostic barriers, and low treatment continuity. Additionally, territorial heterogeneity overlaps with areas affected by conflict, socioeconomic inequality, and displacement, shaping the distribution of mental disorders [25,49,50].

## 5. Conclusions

In Colombia, mental and behavioral disorders remain a major contributor to the national burden of disease. The distribution of this burden shows marked differences by sex and territory, with a higher proportion of YLDs among women and a higher-level occurrence of YLLs among men, reflecting persistent inequalities in social determinants and access to mental health services.

These findings highlight the need to strengthen intersectoral policies aimed at prevention, early detection, and comprehensive care, incorporating territorial and equity-based approaches. Integrating this evidence into health planning and resource allocation could enhance the efficiency of interventions and the responsiveness of the health system. The results may guide national and subnational strategies to optimize resource distribution and strengthen the health system’s response to MBDs in the country.

### Limitations

Limitations inherent to secondary data analysis are acknowledged. The quality of estimates depends on the accuracy and completeness of national records, which may lead to diagnostic errors, underreporting, or duplication of cases. Mortality associated with mental disorders is likely underestimated, as death certificates often prioritize the immediate cause rather than underlying psychiatric conditions. Additionally, regional disparities in diagnostic capacity and the exclusion of categories without assigned weights may limit comparability with international studies. Furthermore, subnational comparisons should be interpreted with caution, as the estimates were not age-standardized, which may affect comparability between regions with different demographic structures.

## Figures and Tables

**Figure 1 ijerph-22-01854-f001:**
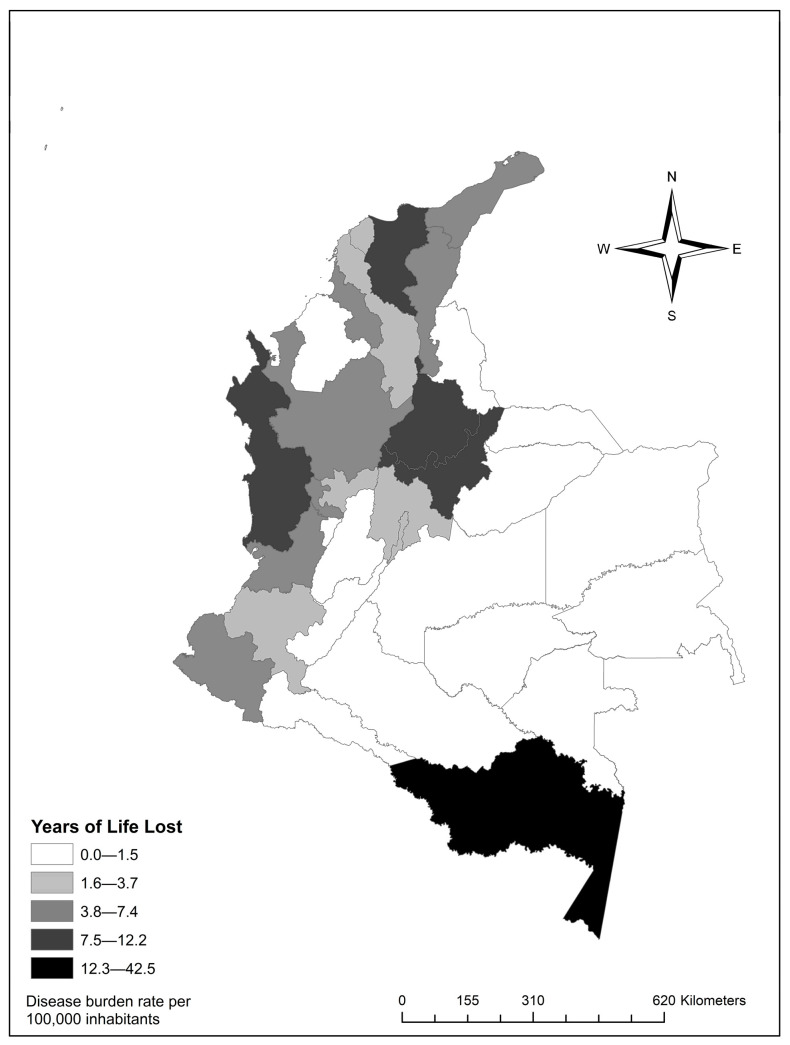
Distribution of Years of Life Lost (YLLs) rates by department.

**Figure 2 ijerph-22-01854-f002:**
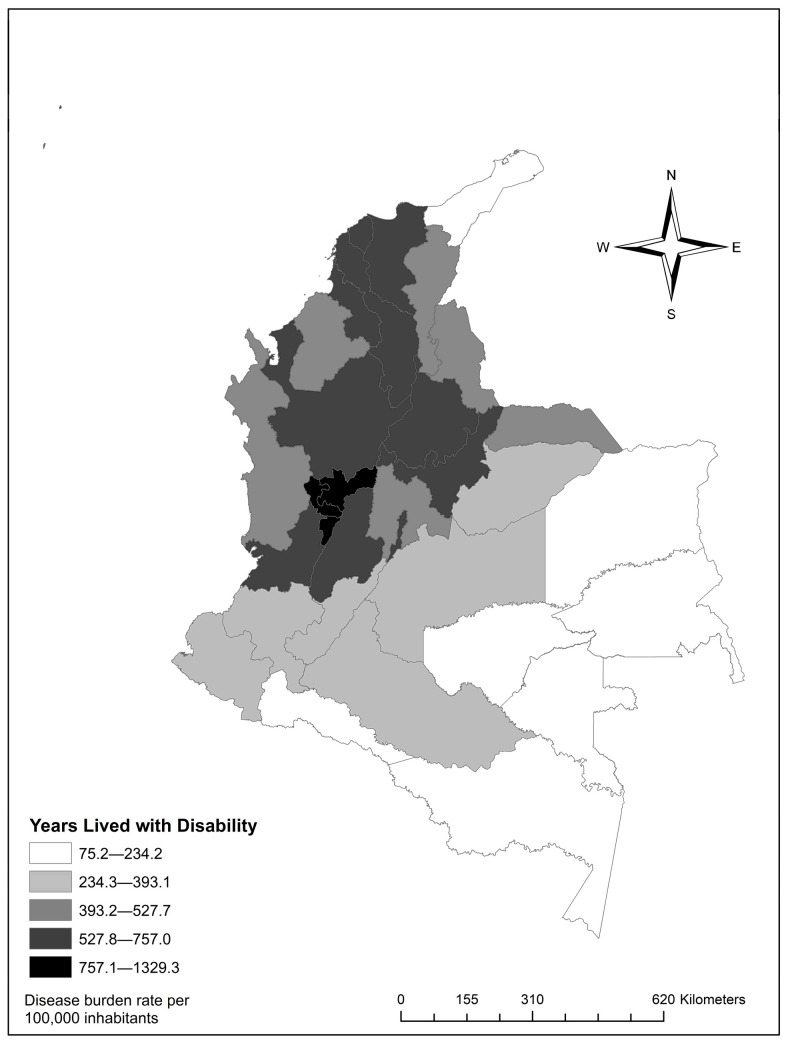
Distribution of Years Lived with Disability (YLDs) rates by department.

**Figure 3 ijerph-22-01854-f003:**
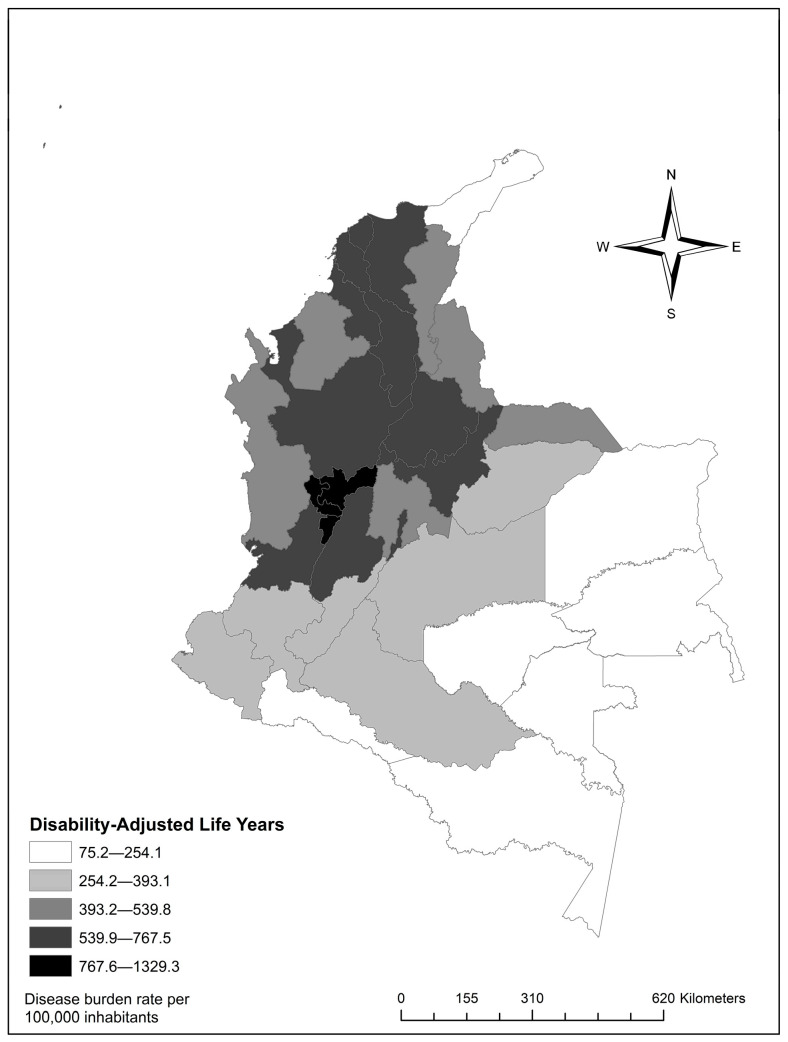
Distribution of Disability-Adjusted Life Years rates by department.

**Figure 4 ijerph-22-01854-f004:**
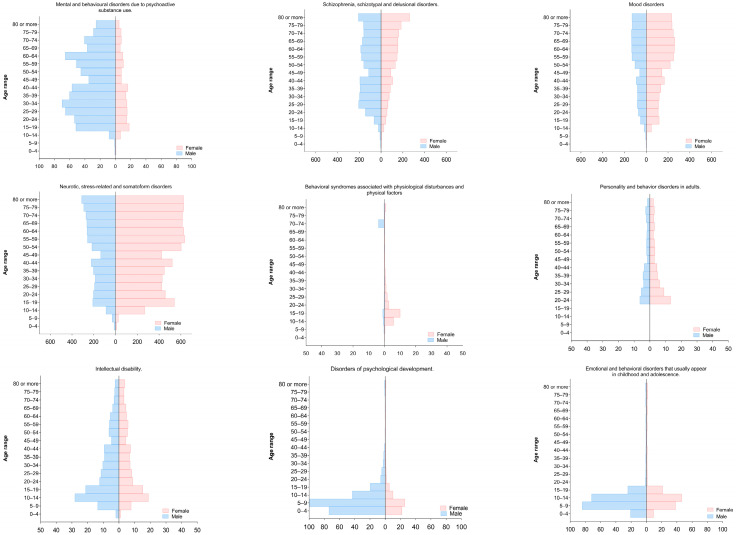
Distribution of disability-adjusted life years by subgroups according to age and sex, 2022.

## Data Availability

The morbidity and mortality data used in this study were obtained from the MSPS (https://www.minsalud.gov.co/Portada/index.html), accessed on 5 March 2024, through SISPRO (https://www.sispro.gov.co/Pages/Home.aspx), accessed on 5 March 2024, which hosts the RUAF and RIPS databases (https://drive.google.com/drive/folders/1NDSv6yfFnpMpPdhG0jZMko75BnO7op1n?usp=sharing). Access to these datasets is restricted and granted upon formal request to the MSPS, in compliance with Statutory Law 1581 of 2012 [14], which authorizes the use of data for historical, statistical, or scientific purposes. Researchers must request access credentials by contacting the MSPS at (sispro_bodega@minsalud.gov.co), and are subsequently provided with login information to access the SISPRO server through an SQL Server Analysis Services cube in Excel for data extraction and processing. Mortality records were obtained from the RUAF–Non-Fetal Mortality database, and morbidity data from RIPS. Demographic information was drawn from the 2022 national population projections published by the National Administrative Department of Statistics (DANE) (https://www.dane.gov.co/index.php), accessed on 6 March 2024.

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
