# Peer review of "Burden of Mental and Behavioral Disorders in Colombia, 2022: A Subnational Analysis Based on Disability-Adjusted Life Years"

_ijerph, 2025, doi:10.3390/ijerph22121854_

Round 1

Reviewer 1 Report

Comments and Suggestions for Authors

The use of the word ‘burden’ is controversial. I am not requesting a term change, but if there is another term that it can be replaced with, I think that would be more appropriate.

Background is sufficient, but could benefit from being a little more concise at times.

Section 2 is good, but needs a little more clarity. I found it challenging to follow.

The use of secondary data analysis could use some rationale.

Figures 1-3  are very effective, I enjoy this take on sharing results visually.

As someone who is unfamiliar with Columbia, I think the results would benefit and be more understood if it listed all the departments/regions in the opening.

The discussion could be more concise and make use of simpler words.

The conclusion needs revisiting to ensure clarity. The writing feels a little more disjointed than the rest of the text. To me, the conclusions are not clear enough.

Your first limitation (L.294-295) would benefit from stating the effect of this. You have stated a potential problem but not its impact.

Overall I liked this paper and feel it qualifies to be published. However, the English could be improved to more clearly express the research and its conclusions more clearly.

Comments on the Quality of English Language

The English could be improved to more clearly express the research and its conclusions more clearly. It could be more concise, and use simpler words to remain clear and avoid ambiguity. 

Reviewer 2 Report

Comments and Suggestions for Authors

This study presents descriptive data based on the national registry and offers important information about the burden of mental disorders. The topic is relevant for many reasons, from public health planning to resource distribution and allocation. Although descriptive registry-based studies are not uncommon, this paper provides new insights from a specific national context. In this context, it is of interest for comparison and for the interpretation of the cross-sectional mental health burden in diverse healthcare systems.

The methods are appropriate. Data source and inclusion criteria are well defined.

I suggest a font size change in Figure 4 to improve the readability of the figure.

The discussion is well written and summarizes the findings.

To strengthen the conclusion, I suggest adding a brief discussion on how these kinds of studies may have implications for public health planning, resource allocation, and new considerations on mental health burden. The references are suitable and updated.

Overall, in my opinion, this paper can add important information on the burden of mental issues, and with these minor revisions, the paper would be enhanced. 

Reviewer 3 Report

Comments and Suggestions for Authors

Dear authors,

I was honoured to read this interesting paper on the burden of mental illness in Colombia. The authors have truly taken on a great task.

Attached are suggestions that, in my opinion, would significantly improve the quality of this paper.

  1. The most serious suggestions relate to the methodology. Such papers require extensive mathematical analysis and transparency in all its phases.
  2. The authors state that they used the “WHO abridged formula” and the “WHO life expectancy table”, but they do not provide the formulas.
  3. It is not clear whether the YLDs were calculated using the incidence or prevalence approach.
  4. There is no information on the disability weights used or which version of the GBD they used. There is no data on the duration of illness by diagnosis, etc. This should be clearly stated in the supplement.
  5. There is no explanation as to whether age-standardized rates by region were used and what the source of the population numbers is.
  6. There is a missing way to group ICD diagnoses. All of this should be in the supplement.
  7. Can you show the “GATHER checklist” in the Supplement?
  8. The introduction is missing information about the Colombian context. Something that would help readers understand the public health context of Colombia.
  9. How do you explain that you have one death per million inhabitants?
  10. All plots need to be explained in detail without reading the text. Are you presenting rates or absolute numbers?
  11. The population pyramids need to be clearer at a better resolution.
  12. Is there a lot of room for deeper analysis?
  13. Secondary data have their limitations, but you have not discussed them in detail. I am afraid that one of the serious limitations is the miscoding and underestimation of mortality in years of life lost.
  14. In the discussion, I would first compare your global results with the results of the GBD study of the University of Washington.
  15. How did you calculate the UI? What statistical software did you use?
  16. If I were you, I would put the Policy Implications and the Linkage to Social Determinants in separate paragraphs.
  17. Results in the abstract without relative rates, only with absolute values.
  18. I would put in the title that a subnational analysis was performed and that Disability-Adjusted Life Years were used. I would present the DALY component in the abstract.
  19. Please double-check the numbers.
  20. Add the contribution of individual authors.

Round 2

Reviewer 3 Report

Comments and Suggestions for Authors

The manuscript does not clearly explain how prevalence was derived from RIPS data. RIPS is a predmoninaltly service-utilization database rather than an epidemiological source. Without detailing case definition, duplicate cleaning, and prevalence estimation, the validity of values is unclear.

Subnational comparisons across Colombian regions are unreliable without age-standardization, since ther is a large demographic differences between them.

Several sections mix YLD values with DALYs, leading to numerical confusion.

Introduction and discussion contain unnecessary repetitions, extensive general information already well known in the DALY literature, and limited linkage to the study’s unique contribution.

Large differences between regions are reported but not adequately analyzed or contextualized.

Frequent grammatical errors, awkward phrases, and unclear sentences reduce readability and may lead to misinterpretation of results.
